# Evaluation of Satisfaction with Healthcare Services in Multimorbid Patients Using PACIC+ Questionnaire: A Cross-Sectional Study

**DOI:** 10.3390/medicina59020276

**Published:** 2023-01-31

**Authors:** Gediminas Urbonas, Gediminas Raila, Danielius Serapinas, Leonas Valius, Džilda Veličkienė, Jurgita Plisienė, Lina Vencevičienė, Elena Jurevičienė, Ida Liseckienė

**Affiliations:** 1Faculty of Medicine, Medical Academy, Lithuanian University of Health Sciences, LT-44307 Kaunas, Lithuania; 2Hospital of Lithuanian University of Health Sciences Kauno Klinikos, LT-50161 Kaunas, Lithuania; 3Institute of Endocrinology, Medical Academy, Lithuanian University of Health Sciences, LT-50009 Kaunas, Lithuania; 4Clinic of Internal Medicine, Family Medicine and Oncology, Faculty of Medicine, Vilnius University, LT-08661 Vilnius, Lithuania; 5Center of Family Medicine, Vilnius University Hospital Santaros Klinikos, LT-08661 Vilnius, Lithuania; 6Faculty of Medicine, Vilnius University, LT-08661 Vilnius, Lithuania

**Keywords:** chronic disease, multimorbidity, patient satisfaction, PACIC+, primary healthcare

## Abstract

*Background and Objectives*: Although multimorbidity poses many challenges for both individuals and healthcare systems, information on how these patients assess the quality of their healthcare is lacking. This study assessed the multimorbid patients’ satisfaction with their healthcare. *Materials and Methods*: This cross-sectional study was a part of a project Joint Action—Chronic Diseases and Promoting Healthy Ageing across the Life Cycle and its implementation. The study included 400 patients with arterial hypertension and at least one concomitant chronic disease. Patients completed The Patient Assessment of Care for Chronic Conditions Plus (PACIC+) questionnaire, EuroQol Five-Dimensions—Three-Level Quality of Life questionnaire, and Hospital Anxiety and Depression scale. *Results*: The mean age of the participants was 65.38 years; there were 52.5% women. The mean PACIC+ 5As summary score was 3.60. With increasing age, participants rated worse on most PACIC+ subscales. Participants who assessed their quality of life as worse were also less satisfied with their healthcare. The presence of three or more concomitant diseases negatively affected PACIC+ scores. Patients with ischemic heart disease and heart failure had lower PACIC+ scores on most subscales, whereas patients with atrial fibrillation had lower scores only on the Agree subscale. The presence of diabetes was not associated with worse PACIC+ scores; moreover, the scores in Assist and Arrange subscales were even better in diabetic patients (3.36 vs. 2.80, *p* = 0.000 and 3.69 vs. 3.13, *p* = 0.008, respectively). Patients with chronic obstructive pulmonary disease, asthma, and musculoskeletal disorders showed lower PACIC+ scores. *Conclusions*: Older age, worse self-assessed health state, presence of three or more diseases, and certain chronic diseases were associated with lower patients’ satisfaction with their healthcare. Personalized healthcare, increasing competencies of primary healthcare teams, healthcare services accessibility, and financial motivation of healthcare providers may increase multimorbid patients’ satisfaction with their healthcare.

## 1. Introduction

Multimorbidity is defined as the co-occurrence of multiple (most commonly, two or more) long-term medical conditions in the same individual. It affects a substantial proportion of ageing societies [1,2]. The prevalence of multimorbidity varies from 20–30% in the whole population [3,4] to 55–98% in the elderly [5].

Multimorbidity poses multiple challenges both to an individual and to health and social care systems [1,2]. For an individual, multimorbidity reduces the quality of life, physical functioning, life expectancy, and increases the risk of disability, psychological distress, as well as the risk of adverse drug reactions resulting from polypharmacy [1,4].

The current healthcare services are not adequately designed to meet the needs of patients with multimorbidity [2]. Multimorbid patients often receive fragmented care, leading to inefficient, ineffective, and possibly harmful clinical interventions [6]. For example, applying simultaneously several single disease-focused guidelines, which are based on data from randomized controlled studies in homogeneous patient groups, may result in overburdening the number of medicines, visits to healthcare institutions, rehabilitation procedures, and lifestyle modifications in multimorbid patients [7]. Multimorbidity contributes to significantly higher healthcare utilization, e.g., it increases the expected use of primary and secondary healthcare services and the risk for unplanned potentially preventable hospitalization, emergency department admissions, and longer hospital stays [8,9].

Multimorbid patients’ healthcare in Lithuania follows global trends. From 2005 to 2019, the proportion of people aged 65 years and over increased from 15.8% to 19.8% [10]. In a large cohort study, based on National Health Insurance Fund data for the period from January 2012 to June 2014, the prevalence of multimorbidity was 42% and 62% at the ages of ≥65 years and ≥85 years, respectively. Furthermore, more than 10% of the population already had at least two chronic conditions at the age of ≥45 years. Cardiovascular diseases (hypertension, ischemic heart disease, heart failure (HF), arrhythmias), diabetes, musculoskeletal diseases (osteoarthritis, back pain), dyslipidemia, stroke, and cancer were the most common chronic conditions accounting for multimorbidity [11]. About 74% of healthcare resources in Lithuania are allocated to multimorbid patients who constitute about 20–30% of the total population [10]. In the cohort study, multimorbidity was associated with an additional 258,761 inpatient days and increased the 30-day re-hospitalization rate by 61%. Outpatient visits in patients with multimorbidity were 2.1-fold more frequent compared with patients having a single disease. Patients with multimorbidity were 9.6-fold more likely to receive home visits by family doctors than patients with a single disease [11].

Multimorbidity research includes not only objective outcomes such as mortality and disability but also subjective health outcomes such as quality of life, well-being and self-rated health [12]. However, because high patient satisfaction is not necessarily equivalent to high quality of healthcare, a broader view is necessary to integrate patients’ subjective views and objective quality indicators into a comprehensive concept of good quality of healthcare [13].

Various measures have been used to assess the quality of care in multimorbid patients. However, the reliance on measures oriented towards a single condition has been a major deficiency [14]. Many measures have also been used to assess the quality of life and functional status in patients with multimorbidity in primary care. While these are particularly valuable for comparing the cost-effectiveness of interventions, they do not measure patient engagement, enablement and empowerment [14].

A patient-centred approach increases patient satisfaction and counters the problems associated with fragmented healthcare, such as contradictory medical advice, overprescribing, over-hospitalization, and unresponsiveness. It requires a coordinated approach to healthcare organization and delivery [15]. The Lithuanian healthcare system faces many issues that may particularly affect multimorbid patients: limited integration of public health and primary healthcare, inadequate coverage of family medicine services, insufficient expansion of outpatient services, insufficient coordination of responsibilities between the different healthcare levels, lack of payment model that incentivizes improvement of healthcare service quality, lack of tools to assess patients’ feedback, and insufficient patients involvement in the process of their treatment [10]. Despite these challenges have been identified and recognized by health authorities, we lack information on how patients self-assess their needs and expectations regarding their morbidity and the healthcare problems they experience. Patient Assessment of Chronic Illness Care (PACIC) is a tool that has been increasingly used in several countries to measure how patients perceive the healthcare they receive.

This study assessed the multimorbid patients’ satisfaction with their healthcare quality and the factors that might affect their satisfaction.

## 2. Materials and Methods

### 2.1. Study Rationale, Design, and Participants

This cross-sectional study was conducted as a part of the EU project Joint Action—Chronic Diseases and Promoting Healthy Ageing across the Life Cycle (JA-CHRODIS) and its implementation project CHRODIS PLUS. The objective of the JA-CHRODIS was to develop a framework for comprehensive care of multimorbid patients potentially applicable across Europe. From 2014 to 2017, an expert consensus has identified sixteen components across five domains (Delivery of Care, Decision Support, Self-Management Support, Information Systems and Technology, and Social and Community Resources) to be addressed for multimorbid patients’ healthcare optimization [6]. As an outcome of this project, an Integrated Care Model for Multimorbidity (ICMM) was developed. Its applicability in clinical practice was assessed within the CHRODIS PLUS project (2017–2020) which included four study sites from Lithuania [16,17].

In Lithuania, CHRODIS PLUS was implemented through the project “Monitoring and evaluation of healthcare for multimorbid patients” for which the approval of Lithuanian Bioethics Committee No L-18-3/3 was received. Here, we present data collected as a part of this project during the ICMM initiation phase in 2018.

The data were collected on patients enrolled at the following CHRODIS PLUS study sites: The Center of Family Medicine of Vilnius University Hospital Santaros Klinikos (Vilnius, Lithuania), private primary healthcare centre (PHC) “InMedica” (Vilnius, Lithuania), The Department of Family Medicine of the Hospital of Lithuanian University of Health Sciences Kauno Klinikos (Kaunas, Lithuania), and Kaltinėnai PHC (Šilalės district, Lithuania). The selected study sites reflect the different PHC settings available in Lithuania: private and public healthcare centres, the affiliates of university hospitals and autonomous healthcare institutions, operating in the largest cities and in rural areas.

The patients with at least two established chronic conditions were selected from the patients’ registers of the four study sites. At the small study sites (“InMedica” and Kaltinėnai PHC), all patients meeting the inclusion criteria and consenting to participate in the study were invited. At the large study sites (The Center of Family Medicine of Vilnius University Hospital Santaros Klinikos and The Department of Family Medicine of the Hospital of Lithuanian University of Health Sciences Kauno Klinikos), eligible patients were randomly selected from the registers attempting to include at least a few patients of each physician working at the site.

The patients were to meet the following inclusion criteria:Age 40–75 years;Presence of arterial hypertension and at least one of the following diseases:
-Diseases of the circulatory system (angina pectoris [ICD-10 code: I20] chronic ischemic heart disease [IHD, I25], atrial fibrillation and flutter [I48], or HF [I50]);-Endocrine, nutritional, and metabolic diseases (type 2 diabetes mellitus [E11], autoimmune thyroiditis [E06.3], postprocedural endocrine and metabolic complications and disorders, not elsewhere classified [E89]);-Diseases of the respiratory system (other chronic obstructive pulmonary disease [COPD, J44], asthma [J45]);-Diseases of the musculoskeletal system and connective tissue (rheumatoid arthritis with rheumatoid factor [M05], polyosteoarthritis [M15], osteoarthritis of hip [M16], osteoarthritis of the knee [M17], osteoarthritis of first carpometacarpal joint [M18], other and unspecified osteoarthritis [M19]; osteoporosis with current pathological fraction [M80]; osteoporosis without current pathological fraction [M81]; dorsalgia [M54]);-Diseases of the nervous system (nerve root and plexus disorders [G54], nerve root and plexus compressions in diseases classified elsewhere [G55]).

The above-mentioned conditions were chosen because of their high prevalence, and the burden to the patient and the healthcare system. All the patients had been diagnosed, and the required treatment had been started before they were included in the study; the time from diagnosis and treatment duration were not considered eligibility criteria.

Once eligible patients were identified, they were invited to participate in the study. During the arranged visit to the study site, the patients were given information about the study and signed the written informed consent form, approved by Lithuanian Bioethics Committee (approval No L-18-3/3). Demographic data (age, gender, marital status, education, employment status, residential location (urban or rural area)) and data on concomitant diseases were collected from the medical records or by interviewing the patients. Then the patients were asked to fill in the questionnaires to evaluate their healthcare quality, quality of life, and mental health state. The study site staff (usually, a case manager) were available to answer questions related to the questionnaire’s filling. All assessments were conducted during a single visit.

### 2.2. Study Objectives and Outcome Measures

The objectives of the study were to assess multimorbid patients’ satisfaction with their healthcare quality before the implementation of ICMM and to investigate the factors that might affect their satisfaction.

The study outcome measures included:-The patient’s assessment of their healthcare quality using The Patient Assessment of Care for Chronic Conditions Plus (PACIC+) questionnaire;-The patient’s assessment of their quality of life using the EuroQol Five-Dimensions—Three-Level Quality of Life (EQ-5D-3L) questionnaire;-The patient’s assessment of their mental health state using the Hospital Anxiety and Depression (HAD) scale.

The PACIC+ questionnaire is a tool to evaluate the patients’ perspective on the received healthcare for their chronic diseases [17]. The original PACIC questionnaire collects patient’s reports on the extent to which he/she has received specific actions and care during the past six months that are consistent with various aspects of the planned, proactive, patient- and population-oriented care for chronic disease (Patient activation, Delivery system design/Patient support, Goal setting/Tailoring, Problem-solving/Contextual, and Follow-up/Coordination). The questionnaire includes 20 items scored from 1 (almost never) to 5 (almost always) [18]. The PACIC+ questionnaire consists of 26 items: 20 items of the original PACIC questionnaire supplemented by six items derived from behavioural counselling, intended to evaluate self-management support and links to community resources [19]. Based on these additional items, the PACIC+ allows to calculate scores in five subscales known as 5As (Assess, Advise, Agree, Assist, and Arrange). The Assess subscale evaluates the extent to which the patient has been asked about his/her concerns and ideas related to their health habits, treatment plan, and visits to other specialists. The Advise subscale includes questions to evaluate how adequately the patient has received information on ways to improve health, e.g., if the patient has been given a treatment plan copy, written instructions, and explanations of how his/her actions may influence the chronic conditions. The Agree subscale evaluates the patient’s own engagement in the healthcare and includes questions related to the problems with the current treatment, opportunities to choose the treatment, and the goals of treatment. The Assist subscale evaluates how the patient has been empowered to better self-care, i.e., whether his/her values and traditions have been considered, whether he/she has been encouraged to seek support, to plan for difficult periods, and to record progress. Finally, the Arrange subscale is based on questions evaluating the receipt of contact after a visit, recommendations on community programs, referrals to a dietitian or a health educator, information on how work, family, and social environment impact the health condition and help to make plans to get support from these sources [19].

Within the CHRODIS PLUS project, the PACIC+ questionnaire was translated into the Lithuanian language [17].

The EQ-5D-3L is a multi-attribute tool to measure health-related quality of life. It consists of two sections: the EQ-5D-3L descriptive system and the EQ visual analogue scale (EQ VAS). The EQ-5D-3L descriptive system comprises five dimensions: mobility, self-care, usual activities, pain/discomfort, and anxiety/depression. The patient is asked to indicate his/her health state by selecting the most appropriate statement (no problems, some problems, or extreme problems) related to each of the five dimensions. The EQ VAS is a quantitative measure of health outcome that reflects the patient’s own judgement. It records the patient’s self-rated health on a vertical VAS where the endpoints are labelled as “Best imaginable health state” and “Worst imaginable health state”. A higher VAS score reflects a better self-perceived health state [20]. The profiles are summarized by calculating a Level Sum Score (LSS), which adds up the levels on each dimension, treating each level’s conventional label (1, 2, or 3) as if it were a number rather than simply a categorical description. A higher LSS implies more severe problems and thus a worse self-perceived health state [21]. This study used a validated Lithuanian version of the EQ-5D-3L questionnaire.

The HAD scale assesses the severity of anxiety and depression in non-psychiatric patients. It consists of an Anxiety subscale (HADS-A) and a Depression subscale (HADS-D) both containing seven intermingled items. The patient rated each item on a four-point severity scale where 0 indicates no impairment and 3—severe impairment. The scores for HAD-A and HAD-D are calculated separately. Scores 0–7 indicate no depression or anxiety, 8–10 indicate a borderline state, and 11–21 indicate an abnormal state [22,23]. The validity of HAD scale has been shown in many clinical settings including primary care [24]. A validated Lithuanian version of HAD scale was used in this study.

### 2.3. Data Analysis

Qualitative (categorical) variables were presented as frequencies (absolute numbers and percentages), and quantitative (continuous) variables were presented as means, standard deviations (SDs), and 95% confidence intervals (CIs). Quantitative variables were not normally distributed and therefore were analyzed using nonparametric statistics. The Mann–Whitney test was used to assess the differences between the two groups, and the Kruskal–Wallis test with multiple pairwise comparisons was used to assess the differences among more than two groups. Statistical differences were interpreted at a 5% (two-sided) significance level. EQ-5D-3L LSS and VAS scores division was conducted according to the database median. IBM SPSS Statistics 27 was used for statistical data analysis.

## 3. Results

### 3.1. Patient Characteristics

In total, 400 patients were enrolled at four study sites: The Center of Family Medicine of Vilnius University Hospital Santaros Klinikos (*n* = 150), private PHC “InMedica” (*n* = 50), The Department of Family Medicine of the Hospital of Lithuanian University of Health Sciences Kauno Klinikos (*n* = 150), and Kaltinėnai PHC (*n* = 50). Two patients who overall provided less than 50% of the data in the questionnaires were excluded from the analysis. Of the remaining 398 patients, 97%, 75%, and 99% completed PACIC+, EQ-5D-3L, and HAD questionnaires, respectively.

The mean age of study participants was 65.38 (SD, 7.67) years, and those aged 60–69 years accounted for the largest group (Table 1). Gender disposition was quite balanced, with slightly more women than men. More than two-thirds of the participants were married; most were well educated.

### 3.2. Overall PACIC+, EQ-5D-3L, and HAD Scale Results

The results of the PACIC+ questionnaire are presented in Table 2. The mean scores of individual 5As subscales ranged from 2.93 (Arrange) to 3.78 (Advice), with the mean summary score being 3.6 (the maximum possible score is 5).

The results of the EQ-5D-3L and HAD scales are presented in Table 3. With the mean LSS score of 7.43 (the maximum possible score is 15) and VAS score of 61.95 (the maximum possible score is 100), study participants assessed their health-related quality of life as intermediate in general. Although the mean HAD scale scores were quite low (Table 3), 28.7% and 13.7% of participants had borderline or overt anxiety or depression as evidenced by HAD-A and HAD-D scores > 7, respectively.

### 3.3. Relationship between PACIC+ Scores and Patients’ Demographic Characteristics, EQ-5D-3L, HAD Scores, and the Presence of Concomitant Diseases

The tables below show the mean scores of PACIC+ subscales and 5As summary scores by age and gender (Table 4), EQ-5D-3L score (Table 5), and HAD subscale scores (Table 6) groups. With increasing age, participants rated worse on most of the PACIC+ subscales. The mean scores, except those of the Assist subscale, were significantly different among the age groups. The Arrange subscale was scored lowest across all age groups (Table 4). There were no differences in the mean PACIC+ scores between men and women (Table 4).

Participants who assessed their quality of life as worse were also less satisfied with their healthcare. The PACIC+ scores were lower in all subscales for the patients with EQ-5D-3L LSS > 7 and in all subscales except Assist and Arrange for the patients with EQ-5D-3L VAS ≤ 60 (Table 5).

No relationship between the presence of borderline or overt anxiety or depression (HAD-A and HAD-D sores > 7, respectively) and the mean PACIC+ scores was observed (Table 6).

The mean PACIC+ subscale scores and 5As summary score were compared between patients with or without certain diseases in addition to arterial hypertension or multimorbidity defined as the presence of at least three chronic conditions (Table 7).

Lower PACIC+ 5As summary scores were found in patients with angina pectoris [I20] and HF [I50], as well as those with osteoarthritis [M15–M19] compared with patients not suffering from these diseases (Table 7).

In the Assess subscale, patients with angina pectoris [I20], HF [I50], or osteoarthritis [M15–M19] scored lower than patients without these diseases. In the Advise subscale, lower mean scores were found for patients with angina pectoris [I20], HF [I50], COPD [J44], and asthma [J45]. In the Agree subscale, lower scores were observed in patients with angina pectoris [I20], HF [I50], and atrial fibrillation and flutter [I48]. In the Assist subscale, patients with COPD [J44], asthma [J45], and osteoarthritis [M15–M19] scored lower than patients without these diseases. Finally, the Arrange subscale scores were lower in patients with HF [I50], COPD [J44], asthma [J45], and osteoarthritis [M15–M19]. The presence of HF was associated with significantly lower scores for all PACIC+ subscales except Assist, whereas the presence of atrial fibrillation and flutter reduced only the Agree subscale (Table 7). The presence of other comorbidities listed as inclusion criteria had no significant effect on PACIC+ scores.

Patients having at least three chronic diseases reported significantly lower scores in all PACIC+ subscales as well as in 5As summary scores compared with patients with two diseases (Table 7).

## 4. Discussion

The mean PACIC+ questionnaire’s 5As summary score of 3.60 (0.93) in our population of multimorbid patients was relatively high compared with the scores in similar patients in other Western countries. The mean 5As summary scores established within the CHRODIS PLUS project before the implementation of ICMM were 2.91 (0.96) in Andalusia (Spain), 3.38 (0.54) in Aragon (Spain), and 3.17 (1.01) in Rome (Italy) [17]. In American diabetic patients 62% of whom had two or more other chronic diseases, the mean 5As summary score was 3.2 (1.0) [25]. In Greek diabetic patients 39% of whom had associated comorbidities, the mean 5As summary score was 3.1 [26]. In German multimorbid patients with osteoarthritis, the mean 5As summary score was 2.52 (1.1) [27]. Similarly, in German multimorbid diabetic patients, the mean 5As summary score was 2.78 (1.0) in patients not involved in diabetes management programs [28]. In Western European patients with IHD having more than 3.3 comorbidities on average, the mean 5A summary score was 2.75 (95% CI 2.69–2.79) [29].

The higher PACIC+ scores in our study can be explained, at least partially, by relatively frequent outpatient visits in Lithuania. The association between the frequency of visits and a patient’s satisfaction with their healthcare has been demonstrated previously [30,31,32]. However, while the differences in 5As summary scores between Lithuania and Italy, Spain, or Greece (approximately 10 vs. 7, 7, or 3 visits per year, respectively) could be attributed to the higher number of yearly visits and hence the better opportunities for patients to ask questions, receive advice and support, this may not be a valid explanation for differences compared to Germany (approximately 10 visits per year) [33]. Another reason for the higher PACIC+ score in our study population might be the proportion of participants from the outpatient departments affiliated with the university hospitals which had already applied the principles of integrated care for patients with multiple chronic conditions [34] before the formal implementation of ICMM. Furthermore, our study population included a considerable proportion of participants with higher education (46.2%), whereas no more than 12% of participants had higher education in other study sites of the CHRODIS PLUS project [17]. The potentially higher interest in the disease and its treatment options, better self-awareness, the ability to raise questions, and previous experience in setting goals and monitoring progress might have resulted in a proactive seeking of information and support.

Our study found decreasing patient satisfaction with their healthcare with advancing age. This trend was also observed in some [19,27,35] but not all [31,32,36,37,38] studies in patients with various chronic conditions, and in one study a positive correlation between PACIC results and age was observed [18]. Although the age effect demonstrated in our study might be explained by decreasing patients’ cognitive abilities and determination to actively engage in their treatment, negative attitudes of medical personnel, reflecting the stereotypes regarding the older people prevalent in the region [39] might have also played a role.

In contrast to some other studies where women rated better [18,35] or worse [29] than men, we did not find an association between gender and PACIC+ scores. However, this finding is in line with the results of many other studies [19,25,31,32,36,37] and it may indicate that the impact of gender on satisfaction with healthcare is indeed negligible.

The results of our study suggest that the presence of depression has no impact on multimorbid patients’ satisfaction with their healthcare. In the PACIC validation study conducted on 255 patients with various chronic conditions, there were about 20% of patients with depression. The overall PACIC score in depressive patients and the scores for the five dimensions were not different from those in patients with other diseases [18]. In a German study which specifically included primary care patients with major depressive disorder, the mean overall PACIC score was 3.25 (0.79) [40], i.e., similar to the score range of 2.49 to 3.80 reported in patients with other chronic conditions [29]. However, one study in diabetic patients (40% of whom also had arterial hypertension) found that depressive states assessed by Center for Epidemiologic Studies Depression Scale correlated with worse PACIC+ results [26]. Thus, the influence of depression on patients’ satisfaction with their healthcare remains to be further clarified.

In our study, the PACIC+ scores, especially in the Assess, Advice and Agree subscales, were lower in patients reporting worse health-related quality of life. Likewise, higher quality of life (assessed by a 5-item scale: very good, good, neutral/reasonable, poor, very poor) was positively associated with PACIC scores in American patients with various chronic conditions and in Brazilian patients with diabetes [41,42]. Similarly, a study in Bosnia and Herzegovina found a higher overall PACIC score in hypertensive patients with self-perceived excellent health in comparison with those with self-perceived bad health [31]. Self-perceived health in this study was measured by asking a question “What do you think about your health?” with the possibility to choose between three answers (excellent, good, or bad). There was a significant association between excellent health with higher scores on Delivery system design/Decision support, Goal setting/Tailoring, and Follow-up/Coordination subscales. Although PACIC+ subscales are not directly comparable with original PACIC dimensions (PACIC+ reflects the implementation of behavioural-counselling-based care, while PACIC reflects the implementation of original ICMM elements), the self-perception of healthcare in patients with lower quality of life may be particularly affected by the lower extent of information and support in problem solving provided to the patients (Decision support/Advise) and patients’ own engagement in their treatment (Goal setting/Agree).

Our patients with three or more chronic conditions had significantly lower PACIC+ 5As summary scores and individual subscales scores compared with patients having two chronic conditions. In the multicenter study in patients with IHD, assessment of healthcare quality was associated with the number of medical conditions, resulting in a 0.01 decrease in the PACIC score with each additional condition [29]. An Australian study also found a significant association between the number of diseases and patients’ satisfaction with their healthcare, i.e., patients with diabetes and IHD/arterial hypertension had lower PACIC scores than those with diabetes only [38]. A weak positive correlation between PACIC scores and the number of comorbidities was reported in the PACIC validation study [18]. On the contrary, no association between patients’ satisfaction with their healthcare and the number of concomitant diseases was demonstrated in patients with type 2 diabetes [25,32,36] or primary care patients in general [43].

Although patients with more diseases may experience some benefits from more frequent visits and increased opportunities to receive information and help, they may also be more sensitive to healthcare fragmentation issues. Recent research has indicated that patients’ satisfaction is probably slightly higher in nurse-led primary care (moderate-certainty evidence), as well as their quality of life may be slightly higher (low-certainty evidence) [44]. There are data supporting the opinion that depression, cognition treatment, and regular exercise can have a positive effect on patients’ satisfaction with their health [45,46].

Therefore, we advocate strengthening the role of a PHC team consisting of a general practitioner, a nurse or broad-scope nurse, a physiotherapist, and a lifestyle medicine specialist, and case management, developing the team’s competencies, and providing better testing possibilities. In addition, it is very important to ensure decision support systems for the PHC team members in the development of telemedicine services and remote general practitioner’s/specialist physician’s consultations.

We found that multimorbid patients’ satisfaction with healthcare depends on the presence of a specific comorbidity. In our study, hypertensive patients with IHD and HF showed lower scores on the Assess, Advise, and Agree subscales compared to patients without these diseases. Likewise, in the large multicentre study in Western European patients with IHD [29], the Assess and Advise subscales, representing the level of gathering and providing information, rated highest, and the Arrange, representing organizational aspects of care, rated lowest. However, the score for the Agree subscale, representing the patient’s engagement, was highest in the above-mentioned study (2.98), while it was among the lowest (2.76) in our study. This may indicate that our patients with IHD may have received less endorsement to be actively involved in their own treatment. HF is a long-term debilitating disease that severely affects the quality of life by the need for hospitalizations, reduced ability to undertake daily living activities, and disturbed psychosocial well-being. The limitations in patients with HF may be even more expressed compared to those with other chronic conditions such as diabetes, cancer, or Alzheimer’s disease [47]. This may have translated into lower satisfaction with healthcare in HF patients. To address these difficulties, The Health Ministry of Lithuania has issued a regulation establishing the role of a specialized HF nurse [48]. When fully implemented, this program could significantly improve the healthcare of HF patients.

Patients with atrial fibrillation reported similar PACIC+ scores as patients without this comorbidity, except for the lower score on the Agree subscale. Studies evaluating the influence of atrial fibrillation on patients’ satisfaction with their healthcare using PACIC or PACIC+ questionnaires are lacking. European Patient Survey in Atrial Fibrillation (EUPS-AF), conducted prior to the approval and widespread uptake of direct thrombin and factor Xa inhibitors, found high patients’ satisfaction with their care: 85.5% of patients rated the quality of care on a five-point Likert scale as good, very good, or excellent [49]. A recent study demonstrated mixed effects of atrial fibrillation treatment choice, with higher satisfaction in patients on direct oral anticoagulants in unadjusted analysis and higher satisfaction in patients on vitamin K antagonists in covariate-adjusted analysis [50]. In Lithuania, most patients with atrial fibrillation use warfarin; therefore, they have to visit a PHC centre for coagulation tests once a month. Moreover, PHC institutions receive an incentive payment for coagulation assessments. More frequent visits might have resulted in better overall satisfaction with healthcare services, but the associated inconvenience might have negatively affected the Agree subscale. In addition, in Lithuania, direct oral anticoagulants are reimbursed only in patients with atrial fibrillation and two (in men) or three (in women) stroke risk factors [51]. Limited opportunity to choose medications might have also contributed to the lower score on the Agree subscale.

Unlike patients with cardiovascular comorbidities, patients with diabetes had similar PACIC+ 5As summary scores as well as scores on the Assess, Advice, and Agree subscales compared with patients without diabetes. However, the mean scores on the Assist and Arrange subscales were significantly higher in diabetic patients (*p* < 0.005). In the PACIC validation study, diabetic patients also achieved significantly higher overall PACIC, Goal setting/Tailoring, and Follow-up/Coordination scores [18]. Similarly, the overall PACIC, the Goal setting/Tailoring, and the Follow-up/Coordination subscales resulted in higher scores in osteoarthritic patients with concomitant diabetes compared to patients with osteoarthritis only [27]. During the validation of the Dutch PACIC questionnaire, diabetic patients reported higher satisfaction with structured chronic care in 14 out of the 20 PACIC items compared with patients with COPD. Higher scores were achieved in overall PACIC score and particularly in the Delivery system/Practice Design and Goal setting/Tailoring dimensions [52]. It seems that diabetic patients are likely to receive more assistance with problem solving, treatment personalization, and arrangement of follow-up support. This may be associated with strict diabetes diagnostic and management protocols, availability of diabetes nurse consultations (up to four times per year or up to twenty-four times in those with diabetic foot), presence of educational programs, activity of patients’ organizations, and high overall attention to diabetic patients in Lithuania. The review of National Diabetes Plans in Europe conducted within the CHRODIS project emphasized the importance of political priority and adequate resource allocation for diabetes management. In this review, Lithuania’s policy was recognized as an example of the successful implementation of good practices from other countries and international guidelines, although admitting some limitations due to the lack of intersectoral collaboration [53].

Patients with chronic diseases such as diabetes are continuously engaged with a PHC team, diabetes nurses and other specialists, including endocrinologists who consult patients with diabetes at least once a year. At a PHC level, patients with diabetes have to undergo glycated haemoglobin testing at least every 3 months. Moreover, regular visits and continuing care of these patients are conditioned not only by the competencies and specialization of PHC teams but also by financial incentive schemes (in Lithuania, a PHC institution receives incentive payment for glycated haemoglobin testing and diabetes specialist nurse consultations). Thus, in general, patients with diabetes attend outpatient clinics frequently and possibly due to that they are more likely to rate healthcare more favourably.

Our patients with musculoskeletal disorders showed low PACIC+ scores, especially on the Assess, Assist, and Arrange subscales. The German study in patients with osteoarthritis suggested that the lower Assist and Arrange scores may show the necessity to improve self-management support, including collaborative goal setting between doctors and patients regarding physical activity which is still underused in arthritis care [27]. This may be also applicable to Lithuanian patients since the reimbursement of physical therapy is not sufficient to provide adequate support. Another reason for lower patient satisfaction may be less strict diagnostic and monitoring protocols for musculoskeletal diseases compared with those established for diabetes and cardiovascular diseases.

Patients with COPD or asthma were less satisfied with their healthcare compared with patients without these diseases as shown by lower Advise, Assist, and Arrange subscale scores. In the validation study of the Dutch PACIC version, patients with COPD rated worse than diabetic patients [52]. In a Swiss study, low baseline PACIC+ scores significantly improved after the implementation of ICMM and educational programs [54], suggesting that patients with respiratory system diseases may largely benefit from integrated care.

Possible reasons why patients with COPD, asthma, and HF were less satisfied with their healthcare might be the following: (1) PHC teams are not financially motivated to ensure regular care for patients with the above-mentioned chronic diseases and (2) PHC teams are not sufficiently specialized in continuing care of these diseases (e.g., there are no nurses who specialize in the provision of care for patients with HF and pulmonary diseases). In Lithuania, patients with HF are treated at tertiary healthcare centres providing specialized medical care. However, it is obvious that the accessibility of such services is limited, and these services should be provided at a PHC level. It is likely that an improvement in healthcare services for patients with pulmonary diseases and HF might lead to higher patient satisfaction in the future.

The major strength of this study is that it was the first study to provide information on how multimorbid patients perceive the quality of their healthcare. However, there were several limitations. Most patients were enrolled from the PHC centres associated with university hospitals; a large proportion of study patients were well educated. Further studies including more patients from different types of PHC centres are needed to extend these results to the general population of multimorbid patients in Lithuania. Furthermore, prospective studies assessing the effect of integrated care including general practitioners, nurses, case managers, mental health counsellors, and social workers could provide sound evidence for the directions in optimizing the healthcare of patients with multimorbidity.

## 5. Conclusions

This study provided information on the perception of healthcare in a selected sample of multimorbid patients. The older age, worse quality of life, presence of three or more concomitant diseases, and specific chronic diseases predisposed lower satisfaction with healthcare as assessed by PACIC+ 5As summary and individual subscale scores. Chronic diseases for which continuous care is standard practice (i.e., atrial fibrillation or diabetes) had a neutral or even positive effect on patients’ satisfaction with their healthcare. This highlights the importance of a personalized approach to the healthcare of patients with different chronic diseases. Increasing PHC teams’ competencies and involvement in continuing and regular patient care, healthcare services accessibility, and financial motivation of service providers may increase patients’ satisfaction. Further studies are needed to clarify the role of nurses and other members of a PHC team, as well as the influence of social factors and systems on patients’ satisfaction with their healthcare.

## Figures and Tables

**Table 1 medicina-59-00276-t001:** Demographic characteristics of the study population.

	N	%
**Gender**		
**Women**	209	52.5
**Men**	189	47.5
**Age group**		
**<60**	86	21.6
**60–69**	165	41.5
**≥70**	147	36.9
**Marital status**		
**Divorced**	46	11.6
**Widowed**	55	13.8
**Married/cohabiting**	275	69.1
**Single/never had a partner**	22	5.5
**Education**		
**Primary**	27	6.8
**Secondary**	112	28.1
**Higher**	184	46.2
**Lower than basic**	75	18.9
**Employment status**		
**Employed**	148	37.2
**Unemployed**	9	2.3
**Housewife**	9	2.3
**Retired**	205	51.5
**Other**	27	6.7
**Residential location**		
**Urban area**	346	86.9
**Rural area**	52	13.1

**Table 2 medicina-59-00276-t002:** Overall results of the PACIC+ questionnaire.

	Assess	Advise	Agree	Assist	Arrange	5As
**N**	382	382	385	386	386	377
**Mean (SD) ***	3.73 (0.95)	3.78 (0.94)	3.53 (1.05)	3.49 (0.96)	2.93 (1.18)	3.60 (0.93)
**95% CI**	3.63–3.82	3.68–3.87	3.42–3.63	3.40–3.54	2.81–2.93	3.51–3.69

* *p* < 0.05 for difference across categories; SD, standard deviation; CI, confidence interval.

**Table 3 medicina-59-00276-t003:** Overall results of the EQ-5D-3L and HAD scales.

	N	Mean	SD	95% CI
**EQ-5D-3L LSS score**	300	7.43	1.84	7.22–7.64
**EQ-5D-3L VAS score**	301	61.95	17.11	60.01–63.89
**HAD-A score**	394	5.68	3.63	5.32–6.04
**HAD-D score**	394	4.37	3.05	4.07–4.67

EQ-5D-3L, EuroQol Five-Dimensions—Three-Level Quality of Life; CI, confidence interval; HAD-A, Hospital Anxiety and Depression Anxiety subscale; HAD-D, Hospital Anxiety and Depression Depression subscale; LSS, Level Sum Score; SD, standard deviation; VAS, visual analogue scale.

**Table 4 medicina-59-00276-t004:** The comparison of PACIC+ score according to age and gender. Data are presented as mean (standard deviation).

	Assess	Advise	Agree	Assist	Arrange	5As
**Age (years)**						
**<60**	4.00 (0.75)	3.94 (0.87)	3.80 (0.82)	3.61 (0.93)	3.18 (1.14)	3.81 (0.81)
**60–69**	3.74 (0.96)	3.86 (0.94)	3.58 (1.02)	3.56 (0.97)	3.06 (1.19)	3.66 (0.92)
**≥70**	3.53 (0.96)	3.58 (0.96)	3.27 (1.18)	3.36 (0.95)	2.63 (1.17)	3.39 (0.96)
***p* value**	0.003	0.011	0.005	0.132	0.001	0.006
**Gender**						
**Women**	3.73 (0.98)	3.74 (0.94)	3.53 (1.04)	3.45 (0.99)	2.96 (1.21)	3.58 (0.94)
**Men**	3.71 (0.93)	3.83 (0.94)	3.51 (1.09)	3.55 (0.92)	2.91 (1.16)	3.63 (0.91)
***p* value**	0.809	0.213	0.858	0.668	0.575	0.544

**Table 5 medicina-59-00276-t005:** The comparison of PACIC+ score according to the level of quality of life (EQ-5D-3L). Data are presented as mean (standard deviation).

	Assess	Advise	Agree	Assist	Arrange	5As
**EQ-5D-3L LSS**						
**LSS ≤ 7**	3.94 (0.83)	4.02 (0.87)	3.81 (0.98)	3.68 (0.90)	3.24 (1.14)	3.83 (0.85)
**LSS > 7**	3.61 (0.97)	3.70 (0.95)	3.50 (1.02)	3.41 (1.03)	2.89 (1.16)	3.52 (0.93)
***p* value**	0.003	0.004	0.005	0.020	0.016	0.004
**EQ-5D-3L VAS**						
**VAS > 60**	3.9 (0.87)	4.02 (0.89)	3.86 (0.93)	3.68 (0.88)	3.23 (1.14)	3.86 (0.83)
**VAS ≤ 60**	3.68 (0.94)	3.76 (0.93)	3.51 (1.05)	3.45 (1.03)	2.96 (1.17)	3.55 (0.94)
***p* value**	0.039	0.011	0.002	0.070	0.052	0.005

LSS, Level Sum Score; VAS, visual analogue scale.

**Table 6 medicina-59-00276-t006:** The comparison of PACIC+ score according to the level of anxiety or depression (HAD scale). Data are presented as mean (standard deviation).

	Assess	Advise	Agree	Assist	Arrange	5As
**HAD-A**						
**HAD-A score ≤ 7**	3.72 (0.99)	3.78 (0.96)	3.49 (1.09)	3.50 (0.98)	2.92 (1.20)	3.60 (0.95)
**HAD-A score > 7**	3.75 (0.88)	3.81 (0.90)	3.64 (0.94)	3.51 (0.91)	2.99 (1.16)	3.64 (0.86)
***p* value**	0.837	0.751	0.286	0.984	0.424	0.914
**HAD-D**						
**HAD-D score ≤ 7**	3.72 (0.97)	3.78 (0.95)	3.53 (1.06)	3.50 (0.97)	2.94 (1.21)	3.61 (0.93)
**HAD-D score > 7**	3.74 (0.90)	3.82 (0.88)	3.54 (0.99)	3.50 (0.86)	2.98 (1.01)	3.59 (0.85)
***p* value**	0.82	0.889	0.913	0.967	0.678	0.798

HAD-A, Hospital Anxiety and Depression Anxiety subscale; HAD-D, Hospital Anxiety and Depression Depression subscale.

**Table 7 medicina-59-00276-t007:** The comparison of PACIC+ score by the presence or absence of comorbidity. Data are presented as mean (standard deviation).

Comorbidity (ICD-10 Code)	N (%)	Assess	Advise	Agree	Assist	Arrange	5As
**Diseases of the circulatory system**							
I20	146 (38.8)	3.59 (1.00)	3.67 (0.96)	3.31 (1.14)	3.41 (0.94)	2.76 (1.16)	3.45 (0.96)
No I20	230 (61.2)	3.81 (0.91)	3.85 (0.92)	3.65 (0.99)	3.56 (0.97)	3.05 (1.19)	3.69 (0.89)
*p* value		0.039	0.038	0.005	0.112	0.052	0.012
I50	125 (33.3)	3.58 (0.97)	3.64 (0.89)	3.23 (1.08)	3.39 (0.93)	2.73 (1.17)	3.41 (0.92)
No I50	251 (66.7)	3.79 (0.94)	3.85 (0.96)	3.67(1.02)	3.55 (0.97)	3.04 (1.18)	3.69 (0.92)
*p* value		0.029	0.014	0.000	0.085	0.028	0.003
I48	83 (22.1)	3.65 (1.01)	3.75 (0.99)	3.27 (1.14)	3.46 (0.98)	2.82 (1.33)	3.48 (1.01)
No I48	293 (77.9)	3.75 (0.94)	3.79 (0.93)	3.59 (1.02)	3.51 (0.96)	2.97 (1.15)	3.63 (0.90)
*p* value		0.433	0.935	0.018	0.592	0.282	0.242
**Endocrine, nutritional, and metabolic diseases**							
E11	155 (41.2)	3.74 (0.96)	3.83 (0.91)	3.54 (1.08)	3.69 (0.93)	3.13 (1.18)	3.65 (0.93)
No E11	221 (58.8)	3.71 (0.95)	3.75 (0.96)	3.51 (1.05)	3.36 (0.96)	2.8 (1.17)	3.56 (0.93)
*p* value		0.645	0.357	0.589	0.000	0.008	0.204
**Diseases of the respiratory system**							
J44-J45	91 (24.2)	3.65 (1.07)	3.53 (1.11)	3.35 (1.14)	3.31 (1,01)	2.64 (1,23)	3.43 (1.02)
No J44-J45	285 (75.8)	3.75 (0.92)	3.86 (0.87)	3.58 (1.03)	3.56 (0.94)	3.03 (1.16)	3.66(0.89)
*p* value		0.758	0.01	0.105	0.046	0.007	0.060
**Diseases of the musculoskeletal system and connective tissue**							
M15–M19	110 (29.3)	3.53 (1.03)	3.67 (0.89)	3.41 (0.99)	3.32 (0.92)	2.73 (1.05)	3.46 (0.88)
No M15–M19	266 (70.7)	3.80 (0.91)	3.83 (0.96)	3.57 (1.08)	3.57 (0.97)	3.02 (1.23)	3.66 (0.94)
*p* value		0.025	0.087	0.174	0.028	0.045	0.049
**Presence of 3 or more chronic conditions**							
Yes	327 (82.2)	3.66 (0.97)	3.72 (0.96)	3.44 (1.07)	3.42 (0.96)	2.81 (1.19)	3.53 (0.94)
No	71 (17.8)	4.02 (0.83)	4.06 (0.83)	3.90 (0.95)	3.88 (0.87)	3.50 (1.02)	3.94 (0.81)
*p* value		0.003	0.007	0.001	0.000	0.000	0.001

E11, type 2 diabetes mellitus; I20, angina pectoris; ICD-10, International Classification of Diseases 10th Revision; I50, heart failure; I48, atrial fibrillation and flutter; J44, chronic obstructive pulmonary disease; J45, asthma; M15, polyosteoarthritis; M16, osteoarthritis of hip; M17, osteoarthritis of knee; M18, osteoarthritis of first carpometacarpal joint; M19, other and unspecified osteoarthritis.

## Data Availability

Not applicable.

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
