# Peer review of "Evaluation of Satisfaction with Healthcare Services in Multimorbid Patients Using PACIC+ Questionnaire: A Cross-Sectional Study"

_medicina, 2023, doi:10.3390/medicina59020276_

Round 1

Reviewer 1 Report

Congratulations on your interesting paper on the evaluation of satisfaction with healthcare by patients with multimorbidity.

Please clarify whether all patients in the study started treatment in 2018. If not, when selecting patients to participate in the study, was their treatment history taken into account - was the patient population analysed in terms of length of treatment - the period from diagnosis to initiation of treatment?

With reference to table 1 showing the demographic characteristics of the population, could you please clarify: 1) what is meant by "other" in the education section; 2) in the marital status section, please clarify whether "singles" also refers to people living in informal relationships?

With regard to the section on informed consent, please specify in what form consent to prticpate in the study was obteined (written?).

Author Response

P 1:

Please clarify whether all patients in the study started treatment in 2018. If not, when selecting patients to participate in the study, was their treatment history taken into account - was the patient population analysed in terms of length of treatment - the period from diagnosis to initiation of treatment?

R1: 

All the patients had been diagnosed, and the required treatment had been started before they were included into the study; the time from diagnosis and treatment duration were not considered as eligibility criteria. [original paper changed at lines 167-169]

P2:

With reference to table 1 showing the demographic characteristics of the population, could you please clarify: 1) what is meant by "other" in the education section; 2) in the marital status section, please clarify whether "singles" also refers to people living in informal relationships?

R1:

For the better understanding in Table 1  "Other" education was changed to "Lower than basic" [original article was changed], in "marital status" clarification was made - "single/never had a partner" [original article was changed].

P3:

With regard to the section on informed consent, please specify in what form consent to prticpate in the study was obteined (written?).

R1:

During the arranged visit at the study site, the patients were given information about the study and signed the written informed consent form, approved by Lithuanian Bioethics Committee (approval No L-18-3/3). [original article was changed at lines 171-173]

Reviewer 2 Report

This research is a very interesting research, Services and management of patients with chronic diseases really need good research innovation.

my comment: please add your research ethics number and mention it in the research method

then in the discussion: by finding the result that the more co-morbidities suffered by the patient will get a negatively affected PACIC+ score, it would be better to add what you do if something like this happens, give the researcher's arguments to cover this problem and quote journal sources that used  as well as in other cases.

in Conclusion: Please also convey how the nursing implications are real in this decade and also express hopes for future research

Author Response

P1:

My comment: please add your research ethics number and mention it in the research method

R1:

In Lithuania, CHRODIS PLUS was implemented through the project “Monitoring and evaluation of healthcare for multimorbid patients” for which the approval of Lithuanian Bioethics Committee No L-18-3/3 was received. [original article lines 128-130]

During the arranged visit at the study site, the patients were given information about the study and signed the written informed consent form, approved by Lithuanian Bioethics Committee (approval No L-18-3/3). [original article lines 171-173]

P2:

then in the discussion: by finding the result that the more co-morbidities suffered by the patient will get a negatively affected PACIC+ score, it would be better to add what you do if something like this happens, give the researcher's arguments to cover this problem and quote journal sources that used  as well as in other cases.

R2:

Original article in "Discussion" section was changed at lines 422-434: Text added

"Although patients with more diseases may experience some benefits from more frequent visits and increased opportunities to receive information and help, they may also be more sensitive to healthcare fragmentation issues. Recent research has indicated that patients’ satisfaction is probably slightly higher in nurse-led primary care (moderate-certainty evidence), as well as their quality of life may be slightly higher (low-certainty evidence) (44). There are data supporting opinion that depression, cognition treatment, regular exercises can have a positive effect on patients’ satisfaction with their health (45, 46).

Therefore, we advocate strengthening the role of a PHC team consisting of a general practitioner, a nurse or broad-scope nurse, a physiotherapist, and a lifestyle medicine specialist and a case management, developing team’s competencies, and providing better testing possibilities. In addition, it is very important to ensure decision support systems for the PHC team members in the development of telemedicine services and remote general practitioner’s- specialist physician’s consultations."

P3:

in Conclusion: Please also convey how the nursing implications are real in this decade and also express hopes for future research

R3:

Original article was changed in conclusions section, text added at lines 549-551:  "

"Further studies are needed to clarify the role of nurses and other members of a PHC team, as well as the influence of social factors and systems on patients’ satisfaction with their healthcare."

Reviewer 3 Report

Dear researcher(s), you are addressing an important and meaningful gap. Your paper is well-written, and it has some important results, and if you edit your paper, it can be much more effective. Here some humble suggestions to improve the paper, I would do the following to strengthen the paper. I have enjoyed reading the paper and am looking forward to seeing the paper published. You could increase the effect of your paper with some more recent studies suggested below or any other studies and not using the suggested ones.

Author Response

Point 1: Title: good and you may clearly consider the title with the research methodology. If the title is brief, comprehensive the readers and researchers will be more likely to benefit from it more. However, you do not have to shorten it.

R1:

The authors meeting decided make no changes in article Title. If it is a very big problem, please write your suggestions. 

My proposal "Multimorbid patients satisfaction dependence on patient characteristics using PACIC+ questionnaire: a cross-sectional study"  was rejected.

Point 2: the introduction section should explain why the study is needed. The author doesn’t provide a clear picture of the research problem at hand. The importance of creating a program to address this problem should be expanded to fill the knowledge gap.

R2:

Text to original article Introduction section was added at lines 84-95:

Multimorbidity research include not only objective outcomes such as mortality and disability, but also subjective health outcomes such as quality of life, well-being and self-rated health (12). However, because high patient satisfaction is not necessarily equivalent to high quality of healthcare, a broader view is necessary to integrate patients’ subjective views and objective quality indicators into a comprehensive concept of good quality of healthcare (13).

Various measures have been used to assess the quality of care in multimorbid patients. However, the reliance on measures oriented towards a single condition has been a major deficiency (14). Many measures have also been used to assess quality of life and functional status in patients with multimorbidity in primary care. While these are particularly valuable for comparing the cost-effectiveness of interventions, they do not measure patient engagement, enablement and empowerment (14).

Text added at lines 105-110: 

Despite these challenges have been identified and recognized by health authorities, we lack information on how patients self-assess their needs and expectations regarding their morbidity and the healthcare problems they experience. Patient Assessment of Chronic Illness Care (PACIC) is a tool that has been increasingly used in several countries to measure how patients perceive their healthcare they receive. 

Point 4: Result: Please consider selecting and displaying the results of the sample. Because the sample group is mostly in urban area, which may affect the modeling and may affect the research results.

R4:

Data in original article was changed in Table 1. There was a misprint between rural and urban area percent. 

Round 2

Reviewer 3 Report

Thank you for this opportunity to review your work. You are already addressed an important and meaningful gap. This paper is well-written, and it has important results. I have enjoyed reading the paper and am looking forward to seeing the paper published.